# Impact of Coastal Sediments of the Northern Dvina River on Microplastics Inputs to the White and Barents Seas

Artyom V. Belesov ⬤, Timofey V. Rezviy ⬤, Sergey A. Pokryshkin, Dmitry E. Lakhmanov, Dmitry G. Chukhchin and Alexandr Yu. Kozhevnikov *⬤

Core Facility Center 'Arktika', Northern (Arctic) Federal University, 163002 Arkhangelsk, Russia
* Correspondence: akozhevnikov@mail.ru

**Abstract:** The Northern Dvina River flowing into the White Sea may be one of the main sources of microplastic (MP) pollution in the Arctic region. The coastal sediments of the Northern Dvina River act as an intermediate link in the transport of microplastics to the areas of the White and Barents Seas. The µFT-IR and Py-GC/MS methods were used to determine that up to 200 particles or 120 mg of MP per kg could accumulate in the coastal sediments of the Northern Dvina River. Coastal sediments tend to accumulate ABS and PS plastic particles with a particle size of around 200 µm. The accumulated microplastics (218 particles or 117 mg per kg of sediment per year) are carried away by strong currents, especially during spring flooding, resulting in pollution of the Barents and White Seas. The obtained data play an important role in assessing the MP pollution of the Arctic region, especially the White and Barents Seas.

**Keywords:** microplastics; FTIR-spectroscopy; GC-MS; sediments

## 1. Introduction

Three hundred and eighty million tons of plastic per year are produced all over the world, and approximately 4.4 to 12.7 million tons of them are passed into the environment [1,2]. The amount of plastic, released into the environment, does not exceed 5% of total production, but plastic is found in soil [3], air [4], water and biota [5,6]. Detected plastic particles can be divided into several groups: macro- (>25 mm), meso- (5–25 mm), micro- (< 5 mm, but more often 500 µm–5 mm, 1–500 µm) [7–10]. The ecological dynamics of microplastic and macroplastic particle accumulation in the ocean are well-studied [11]. However, the qualitative and quantitative composition, and the main sources of microplastics from rivers and coastal sediments are less studied.

Rivers, which flow through regions with high population density, can probably be one of the major sources of plastic and microplastic pollution in the ocean. Thus, the amount of microplastic particles is directly related to the population [12]. The main sources of microplastics are particles formed after washing clothes [13], abrasion of car tires [14] (up to 50% contamination [15]), and destruction of road markings [16]. Some percentage of microplastic pollution also occurs due to tourist and fishing activities [17]. The source variety results in microplastics flowing into water in the form of fibers, pellets, films, and fragments. At the same time, up to 20% of microplastic pollution might leak from the sewage treatment stations [18]. Coastal sediments can accumulate [19] and later release microplastics, contributing to their migration. Thus, the rapid release of accumulated microplastics because of strong currents or tides might lead to large emissions of microplastics.

The study of microplastics in the coastal sediments of rivers in the Arctic region is particularly interesting. It is related to the higher level of pollution in the Arctic Ocean regions affected by Atlantic waters in comparison with the regions affected by the Great Siberian Rivers plumes. Among large Arctic rivers (Yenisei, Ob, Lena, Kolyma, Pechora)

the Northern Dvina River takes a special place due to the presence of densely populated areas in its delta. The presence of the developed pulp and paper industry, shipbuilding factories, power generation plants, and large river/seaports leads to significant emissions of microplastics [20]. A recent study showed that the Northern Dvina River may be considered a major source of microplastic pollution of the White Sea [21], and therefore, the entire Arctic region. However, the morphological and quantitative composition of microplastic contamination in the coastal sediments of this river has not been studied.

Now, the most common method for the analysis of microplastic content is infrared spectroscopy using an infrared microscope (µFT-IR). It is a non-destructive method, which provides the color, size and shape of particles [22]. In addition to optical methods, thermal analysis methods are also widely used. Pyrolytic gas chromatography with mass-spectrometric detection (Py-GC/MS) provides high accuracy and a low detection limit and is capable of different polymer type detection, and allows to simplify the sample preparation [23–31]. At the same time, the sample is destroyed during the analysis with this method; therefore, no additional information about particle shape and size is able to be obtained [32]. The combined application of µFT-IR and Py-GC/MS methods provides the most complete information on the morphological and quantitative composition of microplastic contamination.

Thus, the purpose of this work is to investigate the morphological and quantitative composition of microplastics in the coastal sediments of the Northern Dvina River. We used µFT-IR and Py-GC/MS methods in this study. Samples of the coastal sediments were picked at the beaches of the Northern Dvina River estuary and in the major urban agglomeration, Arkhangelsk. The number of microplastic particles accumulating per kilogram of sludge formed per year was calculated according to the seasonal variability and formation rate of sediments. We estimated the role of coastal sediments in this area in the microplastic pollution of the White Sea as well as the Arctic region in general.

## 2. Materials and Methods

### 2.1. Sampling

Samples were picked during the period from the 1st to the 4th of November 2021. The air temperature was 2 °C, and there was a low tide and no snowfall. Sampling was conducted at 16 points, following the river flow: three points before the city, six within the city, and seven after the city. The top layer of sand (3 cm) from the area of 0.5 m by 0.5 m (0.25 m²) was sampled for analysis at each sampling point. The location of the sampling points is presented in Table S1 and shown in Figure 1.

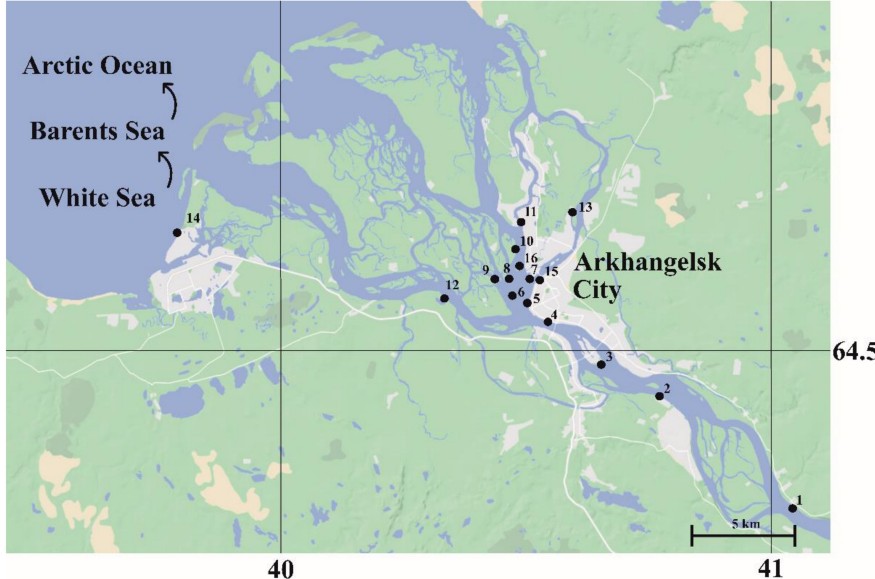

**Figure 1.** Sampling points in the Northern Dvina River.

Orange vests were used during sampling for the estimation of the external pollution of the samples. We used only glass and metal materials during the sampling process and subsequent analysis. Glass jars for sample storage and other metal and glass items were pretreated with isopropanol to exclude the side dopants in the samples.

A field sample was used to monitor sample contamination during sampling and transport. The coastal sediment pre-calcined at 800 °C for removal of the organic component (mostly plastics) was used as a field sample. The field sample was placed near the sampling site in a glass jar with an open lid. After sampling, the lid was closed, and the field sample was transported with the collected samples. The detected contamination of the field sample was taken into account in further calculations.

### 2.2. Sample Extraction and Laboratory Analysis

The subsequent process of sample separation and analysis was carried out under a reverse draft hood. The researchers put on cotton robes while handling samples, to eliminate the possibility of sample contamination. The open Petri dish with a polytetrafluoroethylene (PTFE) filter (Agilent Technologies, Waldbronn, Germany) prewashed with isopropanol, was placed under the hood to control the contamination drift through the air.

For subsequent analysis, the sample portions (30 g for μFT-IR and 10 g for Py-GC/MS) were taken using the five-point method with three parallel measurements. To remove organic impurities, we placed the samples into glass beakers and added Fenton's reagent. Fenton's reagent was prepared with fresh reagents right before the start of the treatment. The mixture was stirred with a glass rod and heated to 70 °C until bubbles appeared, then the heating was stopped, and the beakers were left to cool at room temperature, covered with aluminum foil [33].

For subsequent μFT-IR analysis, the samples were treated by density separation with NaCl solution ($\rho$ = 1.2 g/cm$^3$) according to the method [33]. The supernatant was collected for further sample extraction. The samples were then concentrated on PTFE filters (47 mm diameter, 0.45 μm pore size). PTFE filters were used due to their low cost and the absence of many bands in IR spectra that could overlap the characteristic absorption bands of plastics. During the salt separation, a small amount of sediment was passed onto the filter, causing no significant effect on the subsequent analysis. Between analyses, the filters with samples were stored in glass Petri dishes to prevent contamination drift.

Salt separation was not performed for the Py-GC/MS analysis of the samples due to the low influence of other sediment components on the determination of plastics. After treatment with Fenton's reagent, the samples were concentrated on a pre-weighed glass fiber filter (Whatman GF/C, GE Healthcare, UK) with a pore size of 1.3 μm. The filters with the precipitate were transferred to glass Petri dishes and dried at 105 °C for 10 h. After that, they were weighed and ground in the MM-400 vibration mill (Retscht, Germany) for 3 min with liquid nitrogen cooling. All items used for sample preparation were prewashed with isopropanol. The sample preparation method was based on the other author's data [24,25,34] with minor modifications conducted for the adaptation to the available equipment (vibration mill capacity, glassware, and filter size).

### 2.3. Plastic Standard Samples

The set of plastics was based on [5,13,17,22,25,30] papers and μFT-IR preliminary investigations of the Northern Dvina shoreline sand samples. Polystyrene (PS) (as 430 μm microparticles suspension) was purchased from Sigma Aldrich (USA), high-density polyethylene (HDPE) (SNOLEN IM59164) and polypropylene (PP) (isotactic, S4-6100) pellets were obtained from Gazprom Neftekhim Salavat, Russia. A polymethyl methacrylate (PMMA) (Plexiglas 7N) sheet was from Plexiglas, Germany. Polyethylene terephthalate (PET) (Rospet-A) pellets were from 'Senege New Polymers Plant', Russia. Acrylonitrile butadiene styrene (ABS) (KY-T ABS 757) pellets were obtained from Keyuan, China. All plastics were unpainted and non-recycled.

To obtain model mixtures of microplastics for Py-GC/MS analysis, the weights of powdered plastics were added to quartz sand calcined at 800 °C to obtain a concentration of microplastics from 2 to 60 mg/kg and 10 mm$^3$ of an internal standard solution. The mixtures were milled similarly to samples of real objects. A sample of river sand, cleaned by heating in an oven at 800 °C, was used as a blank sample (laboratory blank).

### 2.4. µFT-IR Analysis

The quantification of detected particles was conducted taking into account the entire surface of the filter. During microscopic observation, microplastics were counted and classified according to the size of the detected particles. Based on the size, microplastics were grouped into the following sizes: <100 µm, 100–300 µm, and >300 µm. After the quantification of microplastics, the chemical identification of microplastics was performed using µFT-IR, which is considered one of the best tests for determining the polymer composition [35]. This method is used to identify the functional groups of plastic materials. Different types of bond configurations give specific IR spectra that help distinguish the chemical composition of the plastic material.

IR spectra were registered with a Vertex 70v FTIR spectrometer and Hyperion 3000 FTIR microscope (Bruker, Bremen, Germany) equipped with a single channel mercury cadmium telluride detector (MCT) and VIS and IR objectives with ×4 and ×15 magnification, respectively. The following spectra register conditions were set up: a spectral range of 4000–400 cm$^{-1}$, resolution of 4 cm$^{-1}$, and the number of scans was equal to 512. A blank filter spectrum was taken as the background spectrum. Each spectrum was analyzed manually using the "OPUS" software (Bruker, Bremen, Germany). Samples were identified by comparison of their spectra with library spectra from the NIST IR-spectra database and the Sigma Aldrich polymers database. These are high-quality spectral databases from the U.S. National Institute of Standards and Technology and leading chemical manufacturer Sigma Aldrich. We also used the spectra of plastic standard samples obtained using attenuated total reflectance attachment GladiATR (PIKE Technologies, Madison, WI, USA) with a diamond prism.

### 2.5. Py-GC/MS Analysis

We used a Shimadzu QP2010Plus (Shimadzu, Japan) gas chromato-mass spectrometer with a Shimadzu QP2010Plus quadrupole mass detector equipped with an EGA/PY-3030D pyrolizer (Frontier Lab., Koriyama, Japan), to determine the quantitative and chemical composition of microplastics with the Py-GC/MS method.

Anthracene D10 (Sigma-Aldrich, Hamburg, Germany) was used as an internal standard for the pyrolysis process. Chromatograms of the standard reference plastics, their mixtures in different ratios and the plastic spiked samples were obtained. A selection of the target pyrolysis products (target compounds) was based on [24–26,29–31] and obtained chromatograms of the plastics mix, the calcinated sand sample, and the matrix sample were known to be free from the target plastics (based on µFT-IR analysis). Retention time and reference ions were used for the identification of the target compounds. The target ion chromatograms were used for quantitative calculations. The ratios of pyrolysis products' peak area to internal standard peak area were plotted against polymer mass for each of the target pyrolysis products. The R2, detection limit and relative standard deviation were also calculated for each of the target plastics pyrolysis products (Table S2, Figure S1).

Pyrolysis was performed by dropping the sample cup into an oven preheated to 700 °C. Pyrolysis products were blown off by carrier gas flow (helium at 20 mL/min) and accumulated in the LN2-cooled trap. After 2 min of pyrolysis, the cryotrap was heated to the initial temperature of the chromatographic column. The initial column temperature of 30 °C was maintained for 2 min, followed by a rise to 120 °C at a rate of 5 °C/min, then to 250 °C at a rate of 10 °C/min, and to 320 °C at a rate of 20 °C/min. The final temperature was held for 10 min. Carrier gas flow into the column was 1 mL/min, the split flow was

20:1, the injector temperature was 300 °C, the mass detector was in scan mode from 40 to 200 Da at 1000 amu/s, and ionization by electron impact at 70 eV.

## 3. Results and Discussion

Microplastic particles were found in all studied samples. Each sample was analyzed in three parallel measurements. A total of 67 microplastic particles were detected in coastal sediments, their number varied from 2.3 ± 0.7 to 9.0 ± 2.7 particles per sampling point. The abundance of microplastics determined by μFT-IR (particles/kg) and by Py-GC/MS (mg/kg) methods correlate with each other (Figure 2). The error in determining microplastic abundance does not exceed 30% for both methods. The number of detected particles varies from 33 to 217 particles/kg. The microplastics concentration varies from 3 mg/kg to 116 mg/kg.

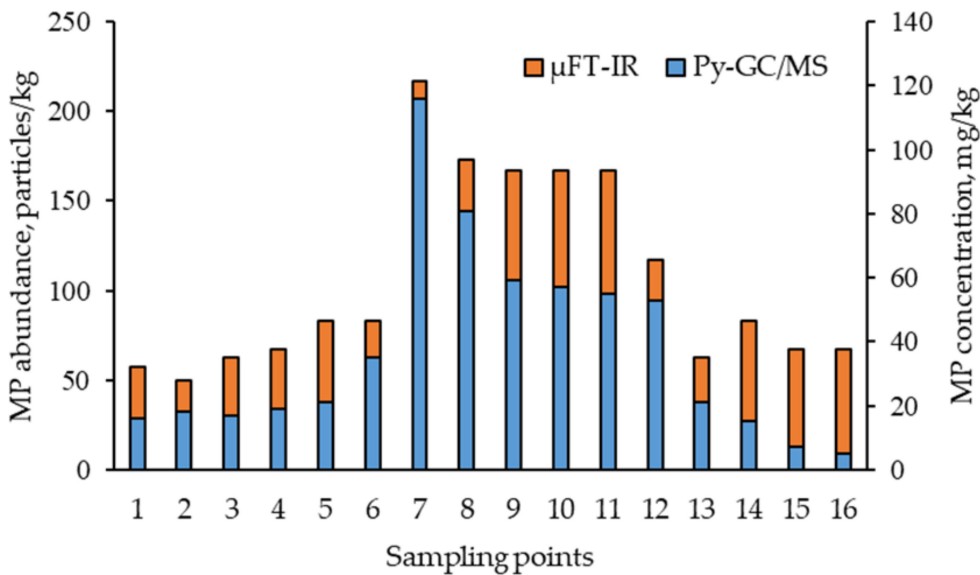

**Figure 2.** The total content of microplastics in the sampling points.

Sampling points 1, 2, and 3 are characterized by the lowest abundance of microplastics, due to their upstream location. The gradual increase in microplastic content at points 4, 5 and 6 can be explained by the nearby proximity to the urban agglomeration. The highest amount of microplastics was found at sampling points 7 and 8 because of the boat dock close to them. Point 14 is located at the city beach of Severodvinsk, the second largest city in the water area of the Northern Dvina, close to the recreation area, which also causes a high content of microplastics in these samples. Points 9, 10, 11, 12, and 13 are located at places where the river begins to separate into branches, leading to strong currents and a high abundance of microplastics. However, the influence of currents can also lead to a decrease in the abundance of microplastics, which is observed for points 15 and 16.

The obtained data about the abundance of microplastic particles correlate with the number of particles found in studies of geographically close areas of the southern Baltic Sea (from 76 to 295 particles/kg [36] and in the coastal zone of the Gulf of Finland (from 15 to 210 particles/kg [37]. One of the main reasons, in this case, is probably the presence of a developed pulp and paper industry, shipbuilding, energy, and large river and sea ports, which, for example, is observed while comparing the number of detected particles in Virginia and North Carolina (600–2200 particles/kg), which are large industrial cities [38] and in the coastal sediments of the urban beach of Cartagena (19–47 particles/kg), which is a developed tourist city [39].

The distance from the center of the urban agglomeration also changes the ratio of the types of microplastics detected. The following types of plastic prevailed among detected particles: Polyethylene (PE), Polypropylene (PP), Polyethylene terephthalate (PET),

Acrylonitrile butadiene styrene (ABS), Polystyrene (PS). Due to the peculiarities of the chemical structure and sample preparation, the IR spectra of PS and ABS are practically indistinguishable, so were combined further (Figures 3 and S2).

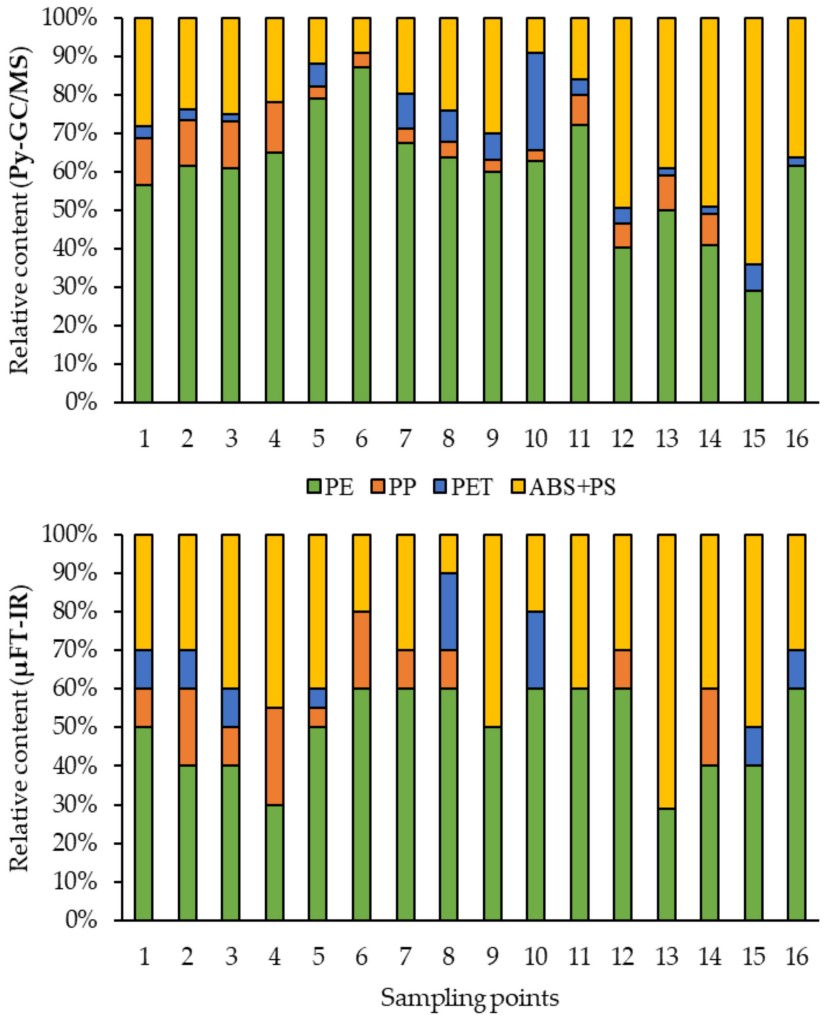

**Figure 3.** Relative content of detected microplastics.

There is an increase in the PE relative content, as we get closer to the center of pollution in the urban agglomeration. Getting away from the center of contamination leads to a predominance of ABS + PS particles. This distribution is probably caused by the higher density of ABS+PS particles compared to PE and, consequently, the more significant influence of currents on PE content. Density separation using NaCl solution (density 1.2 g/cm$^3$) should probably contribute to greater PE and PP extraction in comparison to using ZnCl$_2$ solution (density 1.8 g/cm$^3$) in the case of μFT-IR. However, the obtained data indicate the absence of this correlation, because PE and PP only slightly outnumber PS and ABS. It can be caused by the loss of the smallest particles during the separation and identification process. The absence of density separation and individual particle counting steps allows for comparing the ratio of different microplastic types more precisely in the case of the Py-GC/MS method. It is likely that particles smaller than 100 μm are predominantly PE. This suggestion is indirectly confirmed by the high relative PE content obtained by the Py-GC/MS method.

The difference in the relative content of different types of plastics may also be related to their accumulation in coastal sediments and transport under the action of strong currents, which is confirmed by comparison with the results of previous studies [21]. In the surface water of the Northern Dvina River, the concentration of MP was 3–10 particles per 1000 m$^3$

or 0.01–0.04 mg/m$^3$. According to obtained data, the concentration of MP in coastal sediments was 10–200 particles per 1 kg or 10–50 mg/kg, or six to seven orders of magnitude greater than in surface water. PE and PP were found predominantly among the plastics in surface water (0.94 g/cm$^3$ and 0.91 g/cm$^3$ resp.), which correlates with the obtained data. However, in addition to PE and PP, significant amounts of ABS, PS and PET plastics (1.08, 1.06 and 1.38 g/cm$^3$ resp.) were found in the samples of coastal sediments. These denser MP were absent in the surface water samples [21] but reached up to 70% of the coastal sediments MP pollution. The obtained data allow suggesting, that denser MP particles (ABS, PS and PET) are precipitated into river sediments. This leads to the accumulation of heavy MP in sediments and the formation of MP depot with concentrations higher by six to seven orders of magnitude than in water.

Comparison of the relative MP content in the Arctic seas also indicates the predominance of PE and PP in surface waters and denser plastics (PVC, PTFE, PS) in the subsurface. An important point is the significant pollution of the Barents Sea with microplastics, which correlates with the statement that the Northern Dvina is one of the main sources of plastic pollution of the Barents Sea and the Arctic Sea in general [21,40]. Considering this suggestion, it is important to estimate the volume of total plastic flow from the Northern Dvina River coastal sediments to the Arctic seas.

It is known that there are no significant seasonal fluctuations in the concentration of plastic contamination in the Northern Dvina River [21]. At the same time, the rate of bottom sediment formation in this region per year exceeds 300 g/m$^2$ and is approximately 750 g/m$^2$ [41], which corresponds to 188 g/year, based on the sampling area (0.25 m$^2$). Considering these parameters, we carried out a preliminary estimate of the total plastic flow volume from the Northern Dvina River to the Arctic region (in particular, to the White and Barents Seas) per year (Table 1).

**Table 1.** Estimated number of particles accumulated per year (Values are normalized to mass).

| Value | Sampling Point | | | | | | | | | | | | | | | |
|---|---|---|---|---|---|---|---|---|---|---|---|---|---|---|---|---|
| | 1 | 2 | 3 | 4 | 5 | 6 | 7 | 8 | 9 | 10 | 11 | 12 | 13 | 14 | 15 | 16 |
| Particles/year | 59 | 48 | 64 | 69 | 85 | 85 | 218 | 170 | 165 | 165 | 165 | 117 | 64 | 85 | 69 | 69 |
| Mg/year | 16 | 16 | 16 | 21 | 21 | 37 | 117 | 80 | 59 | 59 | 53 | 53 | 21 | 16 | 5 | 5 |

The calculated number of microplastic particles accumulated in a kg of sediment per year varies from 58 to 218 particles per year. The obtained data significantly exceed the release rate of microplastics during the spring flooding in May (58 pieces/season), as well as during the minimum flow in September (nine pieces/season) [21]. Thus, sediments accumulate microplastic pollution, playing the role of the intermediate stage, and the microplastics are further washed out during the spring flood. However, it should be kept in mind that accumulation at the current stage does not necessarily mean annual particle flux, and the values obtained are only a rough estimate.

The size and morphology of the carried-away microplastic particles are also important characteristics. The data obtained by μFT-IR indicate the predominance of 100–300 μm particles in the composition of coastal sediments. (Figures 4 and S3).

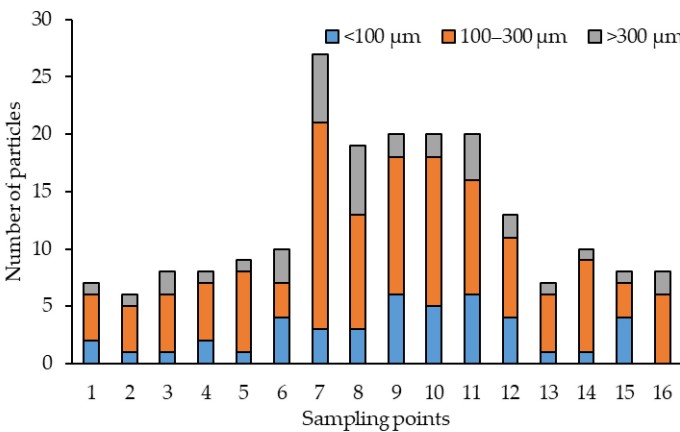

**Figure 4.** The total number of different sized particles depending on the sampling point.

It is expected that there is a gradual increase in particles of all sizes as we move from point 1 to points 6–8. It is caused by approaching the source of microplastic pollution. Even though there is an increase in the relative content of particles smaller than 100 μm particles of 100–300 μm prevail. Possible reasons for this have been described above. It is also likely that particles, smaller than 100 microns, are more easily carried away by the river current, as evidenced by the decrease in their number with distance from the urban area.

In general, particles with different types of morphology (fragments, films, fibers, spheres, foams) were found. The overwhelming majority of particles were fragments (approximately 70%). The particle size varies from 20 to 600 μm. On average the maximum particle size is (419 ± 65) μm and the minimum is (59 ± 15) μm, with an average size of (196 ± 26) μm. More detailed information about the variation in particle size depending on the sampling point is presented in (Figure 5).

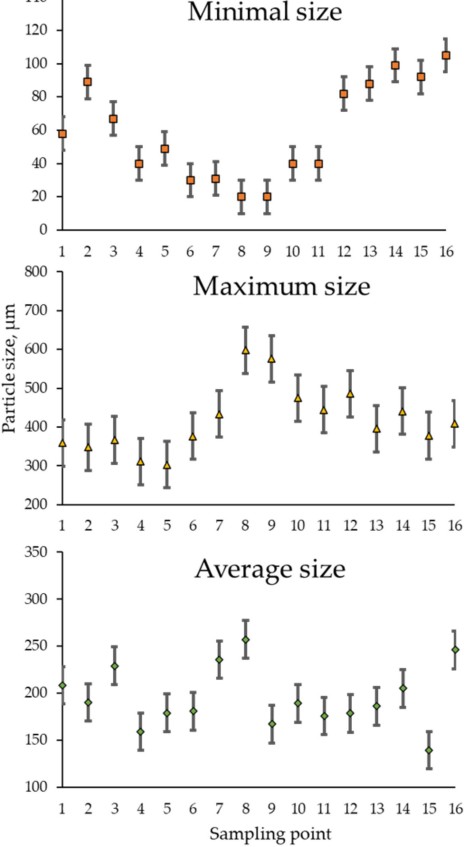

**Figure 5.** Particle sizes depending on the sampling point.

The increase in the particles' total number approaching the urban area (Figure 4) is also accompanied by reaching the minimum and maximum particle size. At the same time, the predominance of 100–300 μm particle sizes leads to an increase in the average particle size. However, the average particle size changes to a lesser degree and lies in the region of 200 μm for most points. The presence of 20 μm particles and a gradual increase in the minimum size confirms the suggestion of gradual drift of the smallest particles and the accumulation of particles larger than 100 μm in coastal sediments.

## 4. Conclusions

The coastal sediments of the Northern Dvina River serve as an important intermediate stage in the ecological dynamics of microplastics carried into the White and Barents Seas. They can probably accumulate up to 218 particles or 117 mg of plastic per year, which can then be released due to strong currents and during spring floods. In some cases, the amount of accumulated microplastics can reach up to 200 particles or 120 mg per kg. Polyethylene prevails among the detected particles (up to 80% of particles). Its content decreases due to the currents with distance from the urban agglomeration.This results in the accumulation (10 to 60% of particles) of denser plastics (ABS + PS) in coastal sediments. The size of these particles can exceed 300 microns, but fragments of microplastics of 200 μm are predominantly accumulated. The obtained data play an important role in the assessment of ecological dynamics of microplastic pollution in the White and Barents Seas, as well as in the Arctic region in general.

**Supplementary Materials:** The following supporting information can be downloaded at: https://www.mdpi.com/article/10.3390/jmse10101485/s1, Figure S1: The pyrogram (in target ions) for the plastic spiked clean (calcinated) sand sample. Figure S2: IR spectra (blue—background, green—library, orange—experimental). Figure S3: Particle samples (A—PS, B—PP, C—ABS). Table S1: Information about each sand sampling point, Table S2: Summary of method performance.

**Author Contributions:** Conceptualization, A.V.B., S.A.P. and A.Y.K.; methodology, A.V.B. and S.A.P.; validation, A.V.B. and S.A.P.; investigation, T.V.R., S.A.P., D.E.L., A.V.B. and D.G.C.; resources, D.G.C. and A.Y.K.; data curation, A.V.B. and S.A.P.; writing—original draft preparation, T.V.R. and D.E.L.; writing—review and editing, A.V.B., S.A.P. and A.Y.K.; visualization, A.V.B. and S.A.P.; supervision, A.Y.K.; project administration, A.Y.K. All authors have read and agreed to the published version of the manuscript.

**Funding:** This research was performed using instrumentation at the Core Facility Center "Arktika" of the Northern (Arctic) Federal University and was supported by the Ministry of Science and Higher Education of the Russian Federation (state assignment project No. 0793-2020-0007).

**Institutional Review Board Statement:** Not applicable.

**Informed Consent Statement:** Not applicable.

**Data Availability Statement:** Not applicable.

**Acknowledgments:** An instrumentation of the Core Facility Center "Arktika" of the Northern (Arctic) Federal University was used in this study.

**Conflicts of Interest:** The authors declare no conflict of interest.

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
