# Peer review of "Impact of Coastal Sediments of the Northern Dvina River on Microplastics Inputs to the White and Barents Seas"

_jmse, doi:10.3390/jmse10101485_

Round 1

Reviewer 1 Report

In this paper, the authors focus on plastic pollutants in the riverine sand sediment in the estuary of the Northern Dvina. The authors use modern technique to quantify the level of plastic pollution in sediments and to study the main characteristics of plastic particles. The paper addresses important processes for estuarine and marine pollution in the Northern Dvina and the White Sea. However, there are certain important drawbacks of the study, which should be considered and reworked within the major revision.

1. In the abstract the authors state that “It has been established those coastal sediments of the Northern Dvina River can act as one of the main sources of microplastic pollution in the Arctic Ocean”. The presented study hardly can support this very significant statement. I recommend to rephrase it and to focus on the White Sea (for which the Northern Dvina is among the major sources of river discharge), because the Arctic Ocean receives huge amount of plastic pollution from the Atlantic Water inflow and discharge from the Great Siberian Rivers (Ob, Yenisei, Lena, etc.). Similarly, I recommend to mention only the White Sea in the last sentence of the abstract.

2. The authors are encouraged to assess the volume of total plastic discharge from the Northern Dvina river to the White Sea. Despite the fact, that they have limited data (only sediment samples), they could make a preliminary assessment based on known velocity of sediment accumulation in the study area as well as seasonal variability of plastic concentration in the Northern Dvina river from previously published papers.

3. Rename the “Discussion” section to “Discussion and Conclusions”. Expand this section based on my previous comment.

Author Response

We are very grateful to the reviewer for his remarks. All comments have been considered and the text has been significantly revised.

  1. The title of the article has been changed. The text of the abstract has been rewritten with a greater focus on the White and Barents Seas. The literature review has also been significantly modified. The possible role of the Northern Dvina River as a source of microplastic pollution of the above seas and the Arctic region as a whole was described in more detail.
  2. The volume of total plastic flow from the Northern Dvina River to the White Sea was estimated based on the rate of accumulation of microplastics in bottom sediments, considering the rate of their formation. The discussion chapter of the results has been considerably expanded.
  3. Due to the expanded chapters with results and their discussion, the "conclusions" paragraph has been left alone.

Reviewer 2 Report

Thank you for the opportunity to review this study.

This manuscript reveals new results to the scientific community in MP analysis and I appreciate the scientific work on the impact of microplastic pollution transfer in the 7 coastal sediments of the Northern Dvina River. However, some parts need to be modified and all paragraphs need more specific details, especially on analytical techniques.  The manuscript needs extensive revision for language and grammar.

The introduction requires significant editing, as it is not written in technical English and cannot be accepted in its current form. In the introduction, there is too low information on the analytical techniques of MicroFTIR and PyrGCMS. Hence, the novelty of this work is not clear, and it is confusing. For instance, other more recent works on the use of these techniques may be inserted and their limitations in MPs analysis should be added.

The main lack of this work is the description of methods used, from the filtration and separation to the quantification and identification of MPs. The details of the procedures used and the explanation of why the authors choose to operate are lacking. The quantification of Micro-FTIR is completely missed (Did they use a counting method? How many particles? Spectra analysis? Libraries? Error of the measures?). Further, reference materials are employed with no details of the typologies or their purchase.  Also, the aim of the use of these materials is not clear.

Regarding the Quality assurance and Quality control part, the authors should insert detailly on the steps of decontamination, the use of glass or steel materials, the reagents used, the field Blancs, potential procedural blanks, the use of gloves, clothes etc… Contamination in microplastics analysis must be evaluated with strict control measures.

Another important part to mention is the total lack of MPs size. The smallest MPs are not mentioned, and they are important to quantify for their potential ingestion by the different organisms and their potential human impact. There is a lack of MPs size in all experimental sections, and I think that it is a significant deficiency of this method, especially if the authors provide a “quantification”.  The average, maximum, or lowest limit of the size should be inserted.

More essential concepts are missing, and I do not think that this article should be accepted in this form. The aims of this method may be interesting as also the implication on the arctic environment, and I appreciate the scientific work, but the authors may clarify better all the paragraphs because I will not recommend that this study has enough quality to warrant publication in its existing form.

Introduction

 The introduction part should focus more on the methods and techniques for MPs analysis, especially for their quantification and identification. The introduction is focused mainly on the sources, the importance of the rivers, and the sampling area. I think that this part requires significant editing to understand the novelty and the aim of the work.

-From the 6° to 12° reference, please add more recent references (2021 or 2022).

-From lines 28 to 30. Please edit the sentence in an English version and use the acronyms (microplastics= MP)

-Lines 32: I do not understand this sentence “but at the same time it has a significant error on the part of the researcher”. Besides one reference is not enough (Pereao et al., 2020), I think it could be better to explain in detail the limitations not errors. For instance, the size limit.

-line 33: “More interesting in terms of quantitative”. It is an opinion. The authors should explain better this sentence with supporting references on methods studies.

-from line 34: English form needs to be modified (for instance use however and not but). All these sentences need to be changed. Further, the sentence is too generic. More details on these techniques are needed.

-from lines 39 to 41. English forms need to completely be modified. 

-from line 39 Why the authors explained the difficulty of MPs studies from the techniques and then to “population”, “sources”, another form of pollution? I think it is complicated to read easily this introduction

-from lines 39 to 49. I think that this part could be inserted before the limitations of the techniques.

Materials and Methods

-Lines 76 English form needs to be modified. Delete “the ingress of plastic”

- QA/QC section and blank control are not considered and more details on the decontamination of materials are needed. For instance, did the authors use a metal sieve? Were they decontaminated? With what? Ethanol? Was the glass beaker decontaminated? Did the authors control the Fenton reagent? Was the PTFE filter cleaned before? Did the authors collect a field blanc? It is not clear. Did the authors prepare a “clean laboratory” with glass and metal materials? More information is needed.

-line 93. The use of salt is not well explained, please add more details on this procedure

-line 94. “A day later” is not scientific and specific. Did the authors use a purification procedure? Recent authors explained that before or during filtration is essential to clean the filter, especially with complex environmental samples. Did the authors have some residues of sand in the filters? Because they could interfere with the analysis, especially with MicroFTIR.

-Lines 103: “A sample of river sand, cleaned by heating in oven at 800°C, was used as a blank  sample.” Which blanc? Procedural or field blanc? Why did the authors use this “clean sand”? If they used it as a field blanc, it is not correct, because the authors did not simulate the contamination during sampling activities.

In general, for the experimental sections: the authors used the reference “39” for the methods of MicroFTIR. Besides this reference is too old (2015) compared to the recent development of Micro-FTIR methods (2021-2022), I did not find any explanation to use this specific method. How do the authors make the quantification? Did they use a counting method? Did they count all the particles on the filter surface? How many particles did they find in the surface?  Did the authors count a specific area of the filter? Also, some best spectra may be inserted in Supplementary materials.  I do not understand how the “quantification of the MPs” were carried out via MicroFTIR. Further, where are the errors in the data? I think that more details and an accurate explanation and comparison of MICROFTIR methods are needed. However, for Pyr-GCMS I did not find any reference on the method used and described by the authors (for instance why 10 g? Did the authors make some tests before?)

-Lines 113: It could be interesting and useful to write the libraries used, maybe in Supplementary materials.

-line 119: “component” is not scientific. The authors should write “chemical composition or chemical typology”

-line 123 “To obtain model mixtures of microplastics, weights of powdered plastics were added …” Which kind of plastics? It is very important to describe detailed the typologies used and why the authors choose them. It is not appropriate to write “model mixture”. The authors should explain better this part. Otherwise, in scientific literature no standard materials for MPs are present, hence why did the authors choose to operate in this way? A better explanation is needed.

-From line 129. How did the authors compare the picks of analytes with reference? Did the authors use libraries? Did the authors compare the reference time with the standard materials inserted? Which kind of plastics? Which picks? The authors should insert this information both in the text and in supplementary materials

Results

-From line 181:

I did not read anything on the size. I think that MPs size is one of the most important factors to determine in this analysis, especially if the authors used a MicroFTIR. The average, maximum and minimum limit of the size (LOD) should be inserted (this is one of the best advantages of the use of MIcroFTIR).

No consideration of size is present and it is unclear this part: ”Thus, large particles of a flat or filamentous shape are more noticeable and the prob-181 ability of their detection by μFT-IR method against the background of smaller particles is 182 higher, which leads to overestimated results. This feature can lead to both an overestimation and an underestimation of the determined relative MP content due to a significant 184 variety of shapes and sizes of the particles under study.” Only in Fig 4 there are three measurements.

I think that a comparison between microFTIR and PyrGCMS is not supported enough.

The Results in general are poor and more detailed info is needed. For instance, besides the size, the authors should insert the errors and more about the shape of MPs. Further, in the Discussion, some limitations of this method presented should be inserted for future studies.  

Author Response

First reviewer

We are very grateful to the reviewer for his remarks. All comments have been considered and the text has been significantly revised.

  1. The title of the article has been changed. The text of the abstract has been rewritten with a greater focus on the White and Barents Seas. The literature review has also been significantly modified. The possible role of the Northern Dvina River as a source of microplastic pollution of the above seas and the Arctic region as a whole was described in more detail.
  2. The volume of total plastic flow from the Northern Dvina River to the White Sea was estimated based on the rate of accumulation of microplastics in bottom sediments, considering the rate of their formation. The discussion chapter of the results has been considerably expanded.
  3. Due to the expanded chapters with results and their discussion, the "conclusions" paragraph has been left alone.

Second reviewer

We are very grateful to the reviewer for his remarks. The proposed revisions significantly improved the quality of the article. All remarks have been considered, and the text has been completely revised.

Abstract

The text of the abstract has been significantly changed, considering the added data and the shift in the focus of the article towards the influence of the Northern Dvina River on the White and Barents Seas.

Introduction

The text of the introduction has been significantly revised. More recent references have been added. The structure of the text has been changed. Significant work has been done on language style. The paragraph about comparing methods has been described more thoroughly. However, this paragraph has not been expanded since the comparison of methods is not the purpose of this article.

Material and methods

Added details for all methods used. Information on quantification of microplastics using infrared spectroscopy has been added. Reference microplastics samples, libraries used, laboratory and field blank samples are described.

Results and discussion

Results obtained using infrared spectroscopy have been added. A comparison of the results of the two methods has been extended. Added description of errors for the obtained data. The volume of total plastic flow from the Northern Dvina River to the White Sea was estimated based on the rate of accumulation of microplastics in bottom sediments, considering the rate of their formation. The discussion of the results is greatly expanded.

Supplementary

Data for PyrGC-MS and IR spectroscopy have been added.

Third reviewer

We are very grateful to the reviewer for his remarks. All comments have been considered and the text has been significantly revised.

The title of the article has been changed to be more specific. Significantly corrected the text of the entire article. Added values of latitude and longitude in Figure 1.

Added explanations about filters used, the Py-GC/MS method, and salt separation, removed discussion about water treatment efficiency.

Reviewer 3 Report

In this paper, authors report levels of microplastics in the coastal sediment samples of the Northern Dvina River. Although many studies have been conducted on the occurrence of microplastics in coastal sediments, this study adds baseline data about the level and potential flux of microplastics to the Arctic Ocean. It could be published in JMSE after major revisions. Specific comments/suggestions are found below:

The title is inappropriate. It does not evaluate the impact of coastal sediments on inputs to the Artic Ocean. The title should be changed more specifically.

Type of the Paper is not specified.

L77: “sand” is inappropriate. It should be changed to “sediment”

In Figure 1, please include latitude and longitude.

L95: Why PTFE filters were used? It is desirable to use metallic filters.

Section 2.4: For Py-GC/MS method, it is required to mention monitoring ions.

L191: The sentence in incorrect. Indeed, it is well-known that wastewater treatment facilities exhibit good efficiency of removing particles including microplastics. The reason why wastewater effluent is important is because of large volume of discharged water.

L214-216: It is likely that density separation using NaCl solution caused greater extraction recoveries for PE and PP. If denser solutions such as ZnCl2 were used, results might be different.

Author Response

(The authors gave the same response as above.)

Round 2

Reviewer 1 Report

The manuscript was improved according to my recommendations and could be published as is.

Author Response

Thank you!

Reviewer 2 Report

Thank you for the authors second reviewer. The  revisions substantially improved the quality of the article and I accept this current form.

Author Response

Thank you!

Reviewer 3 Report

Although the manuscript was improved addressing reviewers’ concerns, it still requires revisions before it is accepted for publication in JMSE.

L13: Numbers do not make sense. It should be presented particles (or mass) per mass of sediment per year.

L25: Further division of microplastics size is unnecessary.

L27-28: This sentence is not needed because this study has nothing to do with ecotoxicity of microplastics.

L67-68: Levels of microplastics should be normalized based on mass of sediment.

Figure 1: Please indicate the name of the city.

L155-156: Please add details about spectra library used. Microplastics detection using FT-IR often depends on the choice of library.

L228-230: What is important in density separation is the density of the solution used, not density of solid crystals. Please revise it.

Table 1 does not make sense. Please revise it. How only 10 particles per year corresponds to Mg per year? Again, values should be normalized per mass. In addition, accumulation at the current stage does not necessarily mean annual flux of particles. Please describe limitations of this very rough estimation.

Author Response

Many thanks to the reviewer for carefully reading our manuscript.

we took into account all the comments on the text of the manuscript. New edits are highlighted in green.